# Syntax-Aware Retrieval Augmented Code Generation

**Xiangyu Zhang**[1]  **Yu Zhou**[1*]  **Guang Yang**[1]  **Taolue Chen**[2*]

[1] Nanjing University of Aeronautics and Astronautics
[2] Birkbeck, University of London

{zhangx1angyu, zhouyu, yang.guang}@nuaa.edu.cn
t.chen@bbk.ac.uk

## Abstract

Neural code generation models are nowadays widely adopted to generate code from natural language descriptions automatically. Recently, pre-trained neural models equipped with token-level retrieval capabilities have exhibited great potentials in neural machine translation. However, applying them directly to code generation experience challenges: the use of the retrieval-based mechanism inevitably introduces extraneous noise to the generation process, resulting in even syntactically incorrect code. Computationally, such models necessitate frequent searches of the cached datastore, which turns out to be time-consuming. To address these issues, we propose $k$NN-TRANX, a token-level retrieval augmented code generation method. $k$NN-TRANX allows for searches in smaller datastores tailored for the code generation task. It leverages syntax constraints for the retrieval of datastores, which reduces the impact of retrieve noise. We evaluate $k$NN-TRANX on two public datasets and the experimental results confirm the effectiveness of our approach.

## 1 Introduction

Neural code generation aims to map the input natural language (NL) to code snippets using deep learning. Due to its great potential to streamline software development, it has garnered significant attentions from both natural language processing and software engineering communities. Various methods have been explored to facilitate code generation (Yin and Neubig, 2018; Wang et al., 2021; Guo et al., 2022). Recent progress in neural machine translation (NMT) shows that the non-parametric $k$-nearest-neighbour machine translation ($k$NN-MT) approach may significantly boost the performance of standard NMT models (Khandelwal et al., 2021) and other text generation models (Kassner and Schütze, 2020; Shuster et al., 2021) by equipping

---

*Corresponding author.

| NL | Check if object *obj* is a string. |
|---|---|
| $k$NN-MT | `all(isinstance(obj)` |
| The correct one | `isinstance(obj, str)` |

Table 1: An example code generated by $k$NN-MT.

the models with a token-level retriever. In particular, this neural-retrieval-in-the-loop approach facilitates the integration of external knowledge into the pre-trained model and provides a simple yet effective method to update the model by switching the retrieval datastore, without fine-tuning the model parameters.

Can such neural-retrieval-in-the-loop approach benefit neural code generation? Our preliminary experiments reveal three main issues (cf. the example in Table 1) if it is adopted outright. Firstly, the model performance may be negatively affected by the noise in the retrieved knowledge. For example, `"all"` does not match the intention of the description, but it is recognized as the target token by the retriever, resulting in the generation of incorrect code. Secondly, the code generated by $k$NN-MT cannot guarantee syntactic correctness, as demonstrated by the mismatching parentheses in the given example. Thirdly, the token-level retrieval method requires similarity search of the entire datastore at each time step of inference, which hinders the deployment of such approach.

In this paper, we propose a novel code generation approach, i.e. $k$NN-TRANX, to overcome the limitations of the neural-retrieval-in-the-loop paradigm for code generation tasks. The basic idea is to integrate symbolic knowledge to ensure the quality of the generated code and expedite the retrieval process. To achieve this, we leverage the sequence-to-tree (seq2tree) model to generate abstract syntax tree (AST), which is a hierarchical tree-like structure used to represent the code, rather than generate target code snippet directly. This enables us to use AST construction rules to guarantee

the syntactic correctness of the generated code and filter out retrieval noise.

We design $k$NN-TRANX as a two-step process (cf. Figure 2). In the first step, we construct two separated datastores, i.e., the syntactic datastore and the semantic datastore, based on the type of AST nodes. This allows us to determine the type of the next node to be predicted according to the grammar rules and query a specific datastore. In the second step, we utilize syntactic rules to filter out irrelevant knowledge and convert the similarity retrieval results of the current target token into a probability distribution, i.e., the $k$NN probability. This probability, together with probability from the neural network, yields the probability of the action to be used for AST generation via a learnable confidence parameter. It can help to minimize retrieval noise and dynamically exploit combinations of the two probabilities, resulting in improved code generation performance.

To evaluate the effectiveness of $k$NN-TRANX, we perform experiments on two publicly available code generation datasets (i.e., CoNaLa and Django). The experimental results show a 27.6% improvement in the exact match metric on the CoNaLa dataset and a 4.2% improvement in the BLEU metric on the Django dataset, surpassing five state-of-the-art models under comparison. Additionally, we conduct an experiment on model canonical incremental adaptation, which updates $k$NN-TRANX by switching the datastore. The experimental results demonstrate that our model can achieve performance comparable to fully fine-tuned models and reduce the trainable parameters by 7,000 times.

## 2 Background

In this section, we provide an overview of the $k$NN-MT paradigm and the seq2tree model.

### 2.1 $k$NN-MT

The $k$NN-MT paradigm (Khandelwal et al., 2021) is a translation mechanism that enhances the quality of model generation by incorporating an additional translation retriever. This allows NMT models to benefit from the retrieved knowledge. The paradigm comprises two main parts, namely, datastore building and model inferring.

**Datastore Building.** The datastore consists of a set of key-value pairs, where the key is the decoder hidden state and the value is the corresponding target token. Formally, given a bilingual sentence pair $(x, y)$ from the training corpus $(\mathcal{X}, \mathcal{Y})$, a pretrained NMT model $f_{NMT}(\cdot)$ generates the $i$-th context representation $h_i = f_{NMT}(x, y_{<i})$, then the datastore $D$ is constructed as follows.

$$D = (K, \ V) = \bigcup_{(x,y)\in(\mathcal{X},\mathcal{Y})} \{(h_i, y_i), \ \forall y_i \in y\}$$

**Model Inferring.** During inference, at time step $i$, given the already generated token $\hat{y}_{<i}$ and the contextual information $\hat{h}_i$, the $k$NN-MT model generates $y_i$ by retrieving the datastore, which can be calculated as

$$p_{k\text{NN}}(y_i \mid x, \ y_{<i}) \propto \sum_{(h_j, y_j)} \mathbb{1}_{y_i = y_j} exp\left(\frac{-d_j}{T}\right)$$

$$(1)$$

where $T$ is the temperature and $d_j$ indicates the $l_2$ distance between query $\hat{h}_i$ and the retrieved key $h_j$.

### 2.2 Seq2tree Model

The purpose of the seq2tree code generation models is to generate ASTs instead of directly outputting code snippets. Compared to the sequence-to-sequence (seq2seq) models, the seq2tree models ensure the syntactic correctness of the generated code. Among the seq2tree models, BertranX (Beau and Crabbé, 2022) was recently proposed and represented the state-of-the-art architecture. BertranX employs BERT to process the input natural language and features a grammar-based decoder.

```
expr = BinOp ( expr left, operator op, expr right )
     | Call ( expr func, expr* args, keyword* keywords )
     | Constant ( constant value)
     | Name ( identifier id, expr_context ctx )
```

Figure 1: Example of ASDL for Python. ASDL defines a set of grammatical symbols, which are denoted in orange and distinguished by a unique constructor name highlighted in blue. Each rule assigns names to its fields or the symbols marked in black. The grammatical symbols can be classified into two types: nonterminals (e.g., expr) and terminals or primitives (e.g., identifier). Some of the grammatical symbols may have qualifiers (*) that allow for zero or more iterations of the symbol.

BertranX describes ASTs using sequences of actions based on ASDL (Wang et al., 1997) grammar, which gives concise notations for describing the abstract syntax of programming languages (cf. Figure 1 as an example). With ASDL, BertranX

defines two distinct types of actions that generate ASTs, i.e., PREDRULE and GENERATE. The first type is used for initiating the generation of a new node from its parent node, which we mark as syntactic nodes in this paper; the second type on the other hand, is used to produce terminal or primitive symbols that we mark as semantic nodes.

## 3  $k$NN-TRANX

The workflow of $k$NN-TRANX is depicted in Figure 2. It consists of two main components: *datastore building* and *model inferring*.

### 3.1  Datastore Building

Given a pre-trained seq2tree model and the training dataset, we first parse code snippets to ASTs and generate all instances in the training corpus. This process allows us to capture and store the decoder representations along with their corresponding target tokens as key-value pairs. The actions that constitute an AST can be categorized into two groups: rules and primitives. These categories align with the actions of GENERATE and PREDRULE, respectively. As shown in Figure 3, the two types of nodes have significant differences in terms of type and quantity. Combining them into the same datastore could potentially reduce retrieval accuracy. Therefore, we employ separated datastores for each node type, referred to as the *syntactic* and *semantic* datastores respectively. Nodes representing the structural information (e.g., *Expr* and *Call*) are put into the syntactic datastore, while nodes representing the semantic information (e.g., *text* and *split*) of the code are put into the semantic one.

Given an NL-code pair $(x, y)$ from the training corpus $(\mathcal{X}, \mathcal{Y})$, we first transform the code snippets $\mathcal{Y}$ into AST representations $\mathcal{Z}$. Next, we calculate the $i$-th context representation $h_i = f_\theta(x, z_{<i})$, where $f_\theta(\cdot)$ refers to the trained seq2tree model and $z \in \mathcal{Z}$. The datastore is constructed by taking $h_i$'s as keys and $z_i$'s as values. Namely,

$$D^{(gra)} = \left( K, \; V^{(gra)} \right)$$
$$= \bigcup_{(x,z) \in (\mathcal{X}, \mathcal{Z})} \{ (h_i, z_i) \mid z_i \in z \; \& \; z_i \in \text{rules} \},$$

and

$$D^{(pri)} = \left( K, \; V^{(pri)} \right)$$
$$= \bigcup_{(x,z) \in (\mathcal{X}, \mathcal{Z})} \{ (h_i, z_i) \mid z_i \in z \; \& \; z_i \in \text{primitives} \}.$$

As a result, two separated symbolic datastores can be constructed based on the various types of target actions within the training set. Constructing datastores in this manner is more effective than storing both types of actions in a single datastore since it helps reduce noise during retrieval. Moreover, the subsequent token type can be determined based on grammar rules, facilitating the retrieval of a specific datastore and accelerating the retrieval process.

### 3.2  Model Inferring

The process of model inference can be divided into three main phases, as shown in Figure 2. First, the code fragment $x$ is put into the trained model to generate the context representation $h_i$ and compute the neural network distribution ($p_{\text{NN}}$). Then, we query the datastore using this representation to obtain the $k$-nearest-neighbor distribution ($p_{k\text{NN}}$). Finally, we combine these two distributions to predict the target token. In the subsequent sections, we will discuss three pivotal components of $k$NN-TRANX: syntax-constrained token-level retrieval, meta-k network, and confidence network.

**Syntax-constrained token-level retrieval.** Given the current context representation $h_i$ generated by the model, we first calculate the $l_2$ distance $d_j = l_2(h_i, \hat{h}_j)$ between the context representation $h_i$ and each neighbor $(\hat{h}_j, \hat{z}_j)$ in the datastore to determine the $k$ nearest neighbors. Previous studies (Meng et al., 2022; Dai et al., 2023) have restricted the search space based on potential input-output patterns to improve decoding efficiency and reduce the impact of noise. However, the restricted search space may also exclude some valuable knowledge.

To mitigate this problem, our approach features syntax-aware retrieval capability. In contrast to conventional seq2seq models, our model aims to generate ASTs that allow to incorporate symbolic knowledge and determine the retrieved tokens by means of syntactic rules. During the retrieval process, we can determine the type of the subsequent token based on the tokens already produced. If the next token is expected to represent the syntactic information, we just retrieve the syntactic datastore to accelerate the retrieval process, and vice versa. Additionally, we can also use the ASDL rules to exclude illegitimate tokens to reduce the amount of irrelevant information. For example, as seen in Figure 2, our model has already generated the

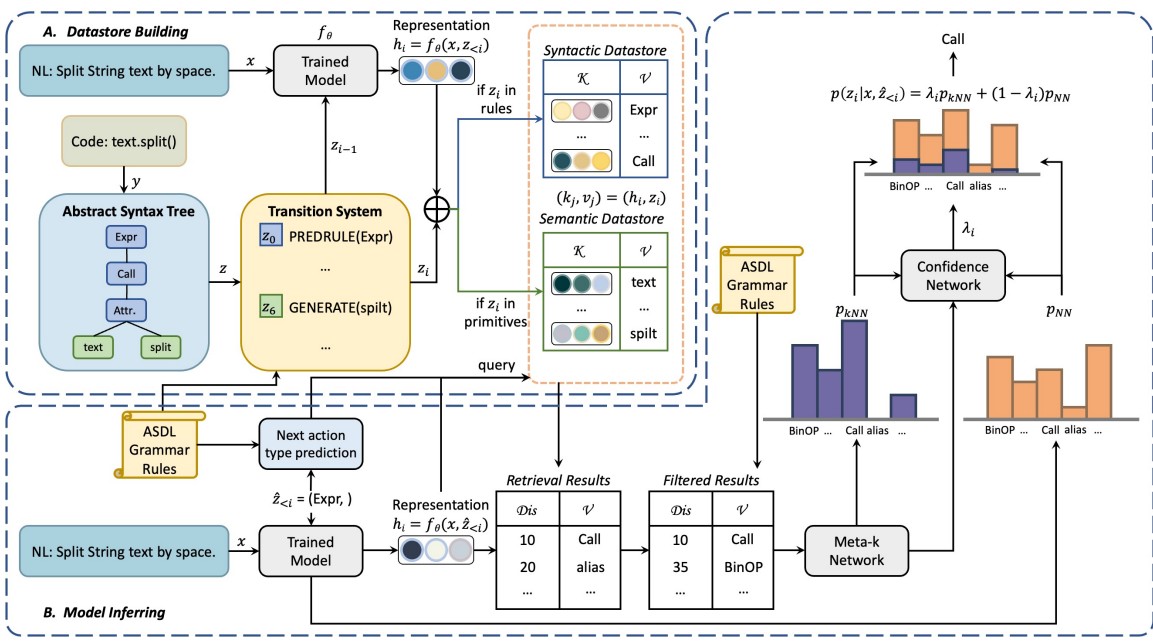

Figure 2: Workflow of $k$NN-TRANX

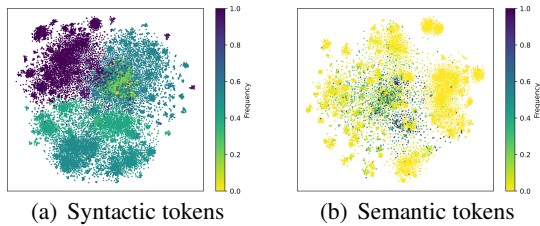

(a) Syntactic tokens      (b) Semantic tokens

Figure 3: t-SNE visualization of CoNaLa data features, with dark colored dots indicating more frequently occurring vocabulary.

node *Expr* in the previous time step. It should be noticed that $k$NN-TRANX have retrieved *Call* and *alias* nodes according to the distances. However, the child nodes of *Expr* do not support *alias* in the ASDL grammar. In this way, we filter out these nodes from the search results to reduce noise and avoid excluding valuable information.

**Meta-k network.** We retrieve $k$ relevant pieces of knowledge from the datastore, and then map the distances between the query vector and the cached representation as probabilities. Empirically, the number of retrievals, $k$, is crucial in our model because too few retrievals may result in valuable information being ignored, while too many retrievals may introduce noise. To alleviate this problem, we employ the meta-k network (Zheng et al., 2021) to dynamically evaluate the weight of the retrieved knowledge. Meta-k network considers a range of values that are smaller than the upper bound $K$, instead of using a fixed value of $k$. Typically

the range is set as $S = \{0, 1, 2, 4, \cdots, 2^{\log_2 \lfloor K \rfloor}\}$. To evaluate the weight of each of the values, we use distance $d_j$ and the count of distinct values in top $j$ neighbors $c_j$ as features and obtain a normalized weight by $p_\beta(k) = \text{softmax}\left(f_\beta([d_1, \ldots, d_K; c_1, \ldots, c_K])\right)$ where $f_\beta(\cdot)$ denotes the Meta-k Network. The prediction of $k$NN can be obtained by $p_{k\text{NN}}(z_i|x, \hat{z}_{<i}) = \sum_{k_r \in S} p_\beta(k_r) \cdot p_{k_r\text{NN}}(z_i|x, \hat{z}_{<i})$ where $p_{k_r\text{NN}}$ indicates the $k_r$-nearest-neighbor prediction results calculated as Equation (1). In this way, the $k$NN model can expand the search space while reducing the impact of the retrieval noise.

**Confidence network.** In order to utilize the knowledge of the symbolic datastore while maintaining the generalization ability of the neural network, we combine two probability distributions by means of weighting. Previous studies have integrated the distributions of $k$NN and NN by using a fixed or learnable parameter $\lambda$ to measure their respective weights. Khandelwal et al. (2021) combine the probability distributions using fixed weights, but this approach fails to dynamically adjust the weights based on the distance retrieved. Zhu et al. (2023) adjust the weights based on the retrieval distance, but they overlook the confidence of the neural network output. Khandelwal et al. (2021) utilize the probability of the neural network and retrieval distance as features to dynamically select the value of $\lambda$ which consider the confidence of

the two distributions, but this approach neglects the correlation between the two distributions.

To address this issue, we propose a confidence network that estimates the confidence of both probabilities based on $k$NN and NN distributions. In addition, we incorporate the weights of each $k$-value as features into the confidence network, ensuring that the model is aware of the number of tokens that require attention. As such our model can capture the relationship between the two distributions. In cases where the two distributions conflict, we assign a higher weight to the distribution with higher confidence. The confidence $\lambda$ is calculated from $\mathbf{W}$ as $\lambda_i = \text{S}(\mathbf{W}[p_{k\text{NN}}(z_i \mid x, \hat{z}_{<i}); p_{\text{NN}}(z_i|x, \hat{z}_{<i}); p_\beta(k)])$, where S denotes the sigmoid activation function.

The final distribution at prediction $z_i$ is calculated as a weighted sum of two distributions with $\lambda_i$, i.e., $p(z_i|x, \hat{z}_{<i}) = \lambda_i p_{k\text{NN}} + (1 - \lambda_i) p_{\text{NN}}$.

## 4 Experiments

In this section, we first introduce the datasets and evaluation metrics. Then, we conduct a comprehensive study and analysis on code generation and model canonical incremental adaptation.

### 4.1 Datasets and evaluation metrics

We evaluate $k$NN-TRANX on two code generation datasets, namely, CoNaLa dataset (Yin et al., 2018) and Django dataset (Oda et al., 2015). The CoNaLa dataset comprises 600k NL-code pairs collected from StackOverflow, out of which 2,879 NL were rewritten by developers. This dataset contains questions that programmers encounter in their real-world projects. On the other hand, the Django dataset consists of 18,805 examples, where each example consists of one line of Python code accompanied by corresponding comments. Compared to CoNaLa, approximately 70% of the examples in Django are simple tasks that include variable assignment, method definition, and exception handling, easily inferred from the corresponding NL predictions. We employed BLEU (Papineni et al., 2002), CodeBLEU (Ren et al., 2020), and exact match (EM) metrics to assess the performance of our experiments.

### 4.2 Code Generation

**Implementation details.** We use BertranX as the base seq2tree model for our experiments, which is trained on annotated data and 100k mined data.

To expedite the implementation, we leverage $k$NN-box (Zhu et al., 2023), an open-source toolkit for building $k$NN-MT, to implement $k$NN-TRANX. As explained in Section 3, $k$NN-TRANX creates two datastores. Due to the considerable difference in vocabulary sizes between the two datastores, we construct separate settings for the syntactic and semantic datastores. For the syntactic datastore, we set the upper limit $K_{rule}$ to 4 to account for its limited token variety. For the semantic datastore, we set $K_{pri}$ to 64. To train the meta-k network and confidence network, we employ the AdamW optimizer with a learning rate of 3e-4. To accelerate the datastore retrieval process, we incorporate the FAISS library (Johnson et al., 2019) for similarity retrieval. All experiments are performed on a single NVIDIA 2080Ti.

**Baselines.** We compare $k$NN-TRANX against five state-of-the-art code generation models.

- **TRANX** (Yin and Neubig, 2018) is a seq2tree model consisting of a bidirectional LSTM encoder for learning the semantic representation and a decoder for outputting a sequence of actions for constructing the tree.
- **Reranker** (Yin and Neubig, 2019) reorders a set of N-best candidates to improve the quality of the generated results.
- **Ext-codegen** (Xu et al., 2020) incorporates API documentation as external knowledge into the model, thus enabling data augmentation.
- **TAE** (Norouzi et al., 2021) uses BERT and a transformer decoder to auto-encoding monolingual data.
- **BertranX** (Beau and Crabbé, 2022) uses BERT as an encoder and serves as the base model for our $k$NN-TRANX.
- **REDCODER** (Parvez et al., 2021) retrieves relevant code from a retrieval database and provides them as a supplement to code generation models.
- **CodeT5** (Wang et al., 2021) builds on the similar architecture of T5 (Raffel et al., 2020) but incorporates code-specific knowledge to endow the model with better code understanding.

**Main results.**

The experimental results are presented in Table 2. Our proposed $k$NN-TRANX exhibits a superior performance over BertranX on the CoNaLa dataset by 3.11 BLEU (9.1%), 2.95 CodeBLEU (8.2%), and 1.6 EM (27.6%). On the Django dataset, we observed improvements of 3.34 BLEU (4.2%), 2.81 CodeBLEU (3.6%), and 2.39 EM (3.0%). These re-

| Model | CoNaLa | | | Django | | |
|---|---|---|---|---|---|---|
| | BLEU | CodeBLEU | EM | BLEU | CodeBLEU | EM |
| TRANX | 28.11 | 29.01 | 2.5 | 62.31 | 64.35 | 73.70 |
| Reranker | 30.11 | 28.98 | 2.8 | 73.26 | 71.29 | 80.18 |
| Ext-codegen | 32.26 | 33.23 | 3.0 | - | - | - |
| TAE | 33.41 | 32.87 | 3.4 | 68.39 | 70.27 | 81.03 |
| BertranX | 34.18 | 36.09 | 5.8 | 79.86 | 78.85 | 79.77 |
| REDCODER | 35.12 | 36.82 | 6.4 | 75.36 | 73.01 | 78.82 |
| CodeT5 | 36.28 | 35.34 | 6.8 | 76.50 | 71.92 | 80.93 |
| $k$NN-TRANX | **37.29** | **39.04** | **7.4** | **83.20** | **81.66** | **82.16** |

Table 2: Comparative results of models trained on the CoNaLa and Django test datasets.

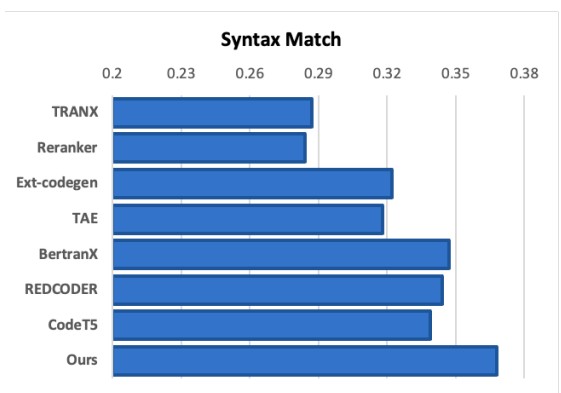

Figure 4: The syntax match score of the generated code on the CoNaLa dataset.

sults indicate that token-level retrieval can improve the performance of code generation models, and the improvement is more evident for challenging tasks. To demonstrate that our model can generate code with more accurate syntax, we provide the syntax match score (Ren et al., 2020) in Figure 4, which reflects the degree of syntax matching in code. The results indicate that our model outperforms the baseline model in terms of syntax accuracy.

**Analysis.** We conduct an ablation study of $k$NN-TRANX as shown in Table 3. The experimental results demonstrate that retrieval filtering method can significantly enhance the performance of the code generation models. For the method that combines NN distribution and $k$NN distribution, we compared the method of measuring retrieval distance proposed in adaptive $k$NN-box (Zhu et al., 2023). Experimental results show that our approach of considering both distributions comprehensively achieves better results. We also evaluate the effect of placing both types of action in the

same datastore. The result shows that this approach significantly reduces the quality of the generated code by introducing a substantial amount of noise. Moreover, we analyze the effect of $K_{rule}$ and $K_{pri}$ on the experimental results, as presented in Table 4. The results align with our conjecture that retrieving a small number of the syntactic nearest neighbors and a relatively large number of semantic entries leads to better code generation. Furthermore, Figure 5 shows that increasing the size of datastore can improve the quality of generated code. Additionally, Figure 5(a) and Figure 5(d) depict that even a small amount of syntactic knowledge stored can achieve a high quality of code generation. In contrast, the quality of generated code keeps improving as the size of semantic datastore increases. We believe this is because semantic knowledge is relatively scarce compared to syntactic knowledge, which can be demonstrated by Figure 3.

### 4.3 Model Canonical Incremental Adaptation

**Implementation details.** Although the recent proposed large-scale models such as Codex (Chen et al., 2021) and GPT-4 (OpenAI, 2023) have demonstrated its powerful code generation capabilities, one of the challenges is that the trained models are difficult to update. Models need to be continuously trained via incremental learning, which consumes huge computing resources. To make matters worse, incremental learning with new data can lead to catastrophic forgetting problems (Li and Hoiem, 2016). To address this, we validate our model using incremental learning by only updating the datastore without adapting the model parameters. We use BertranX[†][1] as our base model to simulate the

---
[1]BertranX[†] is trained on annotated data and 5k mined data on CoNaLa dataset.

| Model | CoNaLa | | | Django | | |
|---|---|---|---|---|---|---|
| | BLEU | CodeBLEU | EM | BLEU | CodeBLEU | EM |
| BertranX | 34.18 | 36.09 | 5.8 | 79.86 | 78.85 | 79.77 |
| $k$NN-TRANX | 37.29 | 39.04 | 7.4 | 83.20 | 81.66 | 82.16 |
| w/o noise filtering | 36.51 | 38.23 | 7.0 | 81.32 | 79.92 | 81.54 |
| w/o Confidence Network | 36.89 | 37.82 | 5.8 | 81.23 | 79.15 | 81.98 |
| w/o separate datastore | 35.62 | 37.21 | 6.2 | 80.74 | 79.23 | 80.08 |

Table 3: Ablation study of different strategies and networks on CoNaLa dataset and Django dataset.

| $K_{rule}$ | $K_{pri}$ | BLEU | CodeBLEU | EM |
|---|---|---|---|---|
| 1 | 1 | 36.05 | 37.74 | 6.4 |
| 1 | 16 | 36.33 | 37.46 | 7.0 |
| 1 | 64 | 36.51 | 37.69 | **7.4** |
| 1 | 128 | 36.63 | 37.84 | 7.2 |
| 4 | 1 | 35.76 | 37.59 | 6.4 |
| 4 | 16 | 37.09 | 38.72 | 7.0 |
| 4 | 64 | 37.29 | **39.04** | **7.4** |
| 4 | 128 | 37.06 | 38.80 | 7.2 |
| 16 | 1 | 36.02 | 37.81 | 6.4 |
| 16 | 16 | 37.01 | 38.70 | 7.2 |
| 16 | 64 | 36.89 | 38.76 | **7.4** |
| 16 | 128 | **37.34** | 38.99 | 7.2 |

Table 4: We studied the impact of different K values on the generation results on CoNaLa dataset. $K_{rule} \in \{1, 4, 16\}$ and $K_{pri} \in \{1, 16, 64, 128\}$.

scenario for incremental learning. Then, we update the datastore using {0k, 10k, 50k, 95k} mined data in CoNaLa dataset respectively. In our experiment, we update the datastore to perform efficient fine-tuning. Compared to the method of fine-tuning all parameters (which requires training 122,205k parameters), our method only needs to train 17k, greatly reducing the GPU memory consumption required for training, and achieving comparable results to fine-tuning all parameters.

It is worth noting that, compared to the previous $k$NN generative model, our model includes two datastores for syntactic and semantic information, respectively. There are 109 kinds of token in the syntactic datastore, and the number of corresponding datastore entries is 1,603k. However, the types of token in the semantic datastore may be infinite, depending on the actual defined vocabulary, so the knowledge corresponding to each vocabulary is relatively scarce, with only 518k entries in this ex-

periment. Figure 5 confirms that providing a small amount of syntactic knowledge can improve the performance of the model. Therefore, we consider two ways to update the datastores in our experiments, i.e., updating both datastores and updating the semantic datastore only.

**Main results.** We adopt the same evaluation metrics as code generation. As shown in Table 5, firstly, without using additional datastore, $k$NN-TRANX[†] can outperform BertranX[†]. As the knowledge in the datastore is constantly updated, we can see that $k$NN-TRANX has improved on three evaluation criteria. In terms of BLEU evaluation, $k$NN-TRANX[†] with 95k external data can achieve performance comparable to that of training-based BertranX. Furthermore, we update only the semantic datastore, which can also be effective. We also provide two examples to demonstrate how $k$-nearest-neighbor retrieval assists in model decision-making in Appendix A.1. It should be noted that the CoNaLa dataset was obtained through data mining, and the majority of the NL-code pairs obtained through mining are irrelevant, which greatly adds noise to our retrieval. Therefore, we believe that $k$NN-TRANX can perform even better on more reliable data sources through incremental learning.

## 5  Related Work

**Code generation.** Code generation aspires to generate target code through natural language to improve programmers' development efficiency. Yin and Neubig (2018) propose TRANX to generate ASTs instead of generating code snippets directly. Based on TRANX, Beau and Crabbé (2022) propose BertranX relying on a BERT encoder and a grammar-based decoder. Poesia et al. (2022) propose a framework for substantially improving the reliability of pre-trained models for code generation. Chen et al. (2022) propose CodeT to leverage pre-trained language models to generate both

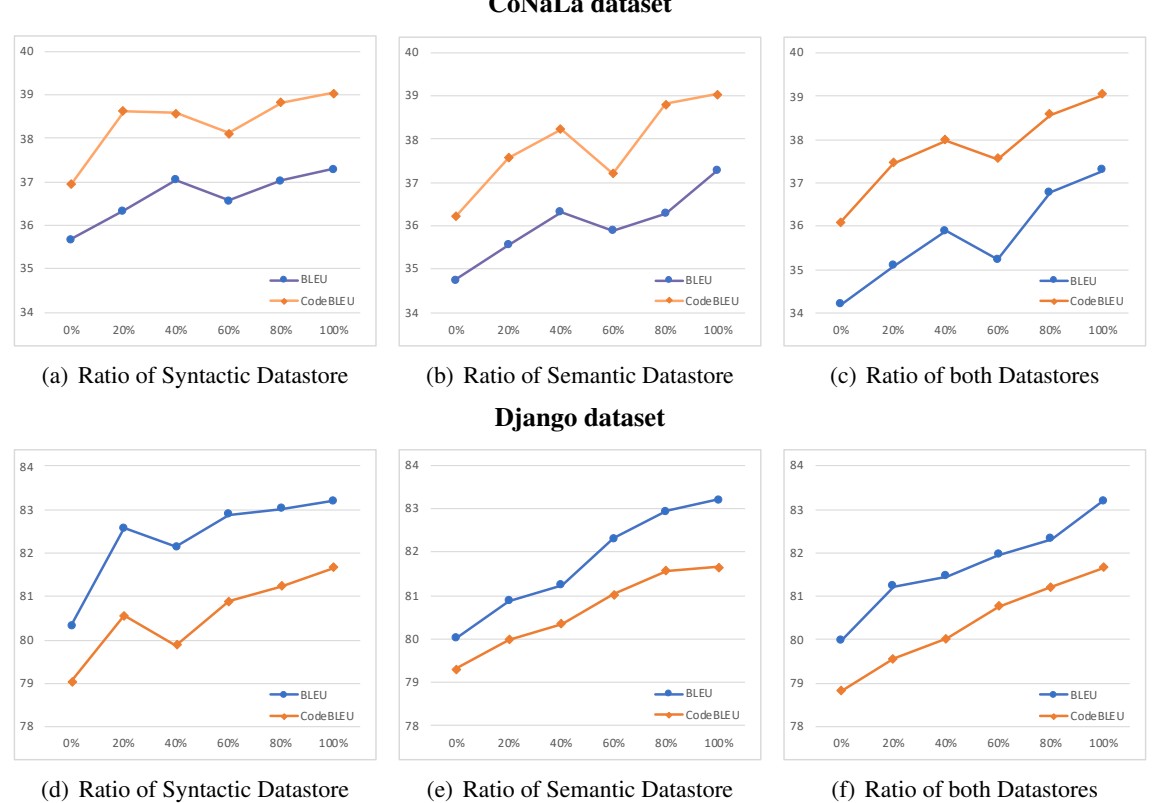

Figure 5: The impact of datastore size on the BLEU and CodeBLEU scores of CoNaLa dataset and Django dataset. We used three strategies to reduce the datastore size, which are reducing the storage of syntactic knowledge, semantic knowledge, and both types of knowledge.

the code snippets along with test cases, and select the best code based on the number of test cases passed. Besides, researchers used pre-training methods to incorporate more external knowledge into the model, effectively improving its performance on downstream tasks (Feng et al., 2020; Wang et al., 2021; Ahmad et al., 2021). Recently, large-scale pre-training models have demonstrated remarkable capabilities in code generation (Li et al., 2023; Wang et al., 2023; Touvron et al., 2023).

**Retrieval-augmented models.** Retrieving and integrating auxiliary sentences has shown effectiveness in enhancing the generation of class models. Farajian et al. (2017) propose retrieving similar sentences from the training set to adapt to different inputs. The work of Hayati et al. (2018) is built on the neural network model of AST driver and generates code by searching action subtrees. Parvez et al. (2021) propose using retrieved natural language code to improve the performance of code generation and code translation models using REDCODER.

Recently, Khandelwal et al. (2021) propose $k$NN-MT, a non-parametric paradigm for construct-

ing a datastore using a decoder representation as a key and using the corresponding target character as a value. The generation is done by retrieving the top $k$ neighbors as the result. Based on this paradigm, a number of optimization methods have also been proposed. Zheng et al. (2021) use a meta-k network to dynamically adjust the retrieval weights. Jiang et al. (2022) improve the robustness of $k$NN-MT in terms of both model structure and training methods. Meng et al. (2022) propose Fast $k$NN-MT, which improves retrieval efficiency by constructing multiple smaller datastores.

## 6 Conclusion

In this paper, we propose $k$NN-TRANX. By providing syntactic and semantic datastores for seq2tree model, we are able to outperform the baselines. In addition, we provide more knowledge for the model by switch the datastores without fine-tuning the neural network. Experimental results show that $k$NN-TRANX exhibits competitive performance against learning-based methods through incremental learning. In the future, we plan to construct a smaller and more fine-grained syntactic datas-

| Model | Only Semantics | External Data | BLEU | CodeBLEU | EM | Training Cost (FLOPs) | Extra Storage |
|---|---|---|---|---|---|---|---|
| BertranX[†] | - | - | 29.22 | 29.89 | 4.0 | $1.09 \cdot 10^{17}$ | - |
| $k$NN-TRANX[†] | - | 0 | 32.25 | 32.08 | 4.2 | $1.11 \cdot 10^{17}$ | 0.2G |
| $k$NN-TRANX[†] | False | 10k | 32.95 | 32.87 | 4.4 | $1.14 \cdot 10^{17}$ | 0.5G |
|  |  | 50k | 33.71 | 33.58 | 4.4 | $1.23 \cdot 10^{17}$ | 1.3G |
|  |  | 95k | 34.04 | 34.69 | 4.8 | $1.35 \cdot 10^{17}$ | 2.2G |
| $k$NN-TRANX[†] | True | 10k | 32.28 | 32.14 | 4.2 | $1.14 \cdot 10^{17}$ | 0.3G |
|  |  | 50k | 33.25 | 33.87 | 4.2 | $1.23 \cdot 10^{17}$ | 0.5G |
|  |  | 95k | 33.89 | 34.43 | 4.6 | $1.35 \cdot 10^{17}$ | 0.8G |
| BertranX | - | - | 34.18 | 36.09 | 5.8 | $1.55 \cdot 10^{18}$ | - |

Table 5: The results of model canonical incremental adaptation. BertranX[†] is trained on cleaned 2k and mined 5k data. $k$NN-TRANX[†] is built on the same size data as a datastore on top of BertranX[†]. When Only Semantics is set to False, both datastores are updated simultaneously, while True means only semantic datastore is updated. External data refers to the quantity of training data updating the datastore.

tore to reduce the search space of the model and accelerate model inference.

## Limitations

Although our proposed approach can enhance the generalizability of the seq2tree model and enable rapid updates by switching datastores, incorporating extra datastores necessitates a similarity search at each time step of inference. Even though only one of the two datastores needs to be retrieved, the inference time for the model may still increase considerably (cf. Appendix A.2). Furthermore, the incorporation of additional datastores will consume storage space and may escalate with growing training data. Although the previous work has proposed approaches to reduce the content of datastore through pruning, this approach also leads to model performance degradation.

## Acknowledgements

This work is supported by the National Natural Science Foundation of China (No. 61972197, No. 62372232), the Natural Science Foundation of Jiangsu Province (No. BK20201292), and the Collaborative Innovation Center of Novel Software Technology and Industrialization. T. Chen is partially supported by an overseas grant from the State Key Laboratory of Novel Software Technology, Nanjing University (KFKT2022A03), Birkbeck BEI School Project (EFFECT) and National Natural Science Foundation of China (No. 62272397).

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

# A   Appendix

## A.1   Case Study on Model Canonical Incremental Adaptation

We showcase two selected instances from the CoNaLa dataset in Table 6. For each instance, we present the output and probability distribution of the model when generating incorrect behavior. We also demonstrate how our method utilizes the $k$NN algorithm to enhance the model's decision-making process, thereby proving the effectiveness of our method. As shown in the first example, BertranX[†] demonstrates the ability to construct code with appropriate syntax, but it inaccurately generates the primitive *legend*. However, with the assistance of $k$NN-TRANX[†], the final decision of the model changes through retrieval, resulting in the desired code. In the second example, BertranX[†] makes an error when generating the second token. $k$NN-TRANX[†], utilizing an updated syntactic datastore, produces code that is semantically close to the target code. In contrast, kNN-TRANX[†], which did not update the syntactic datastore, has drawbacks in the generation of correct code due to insufficient syntactic node information, despite having improved retrieval efficiency and datastore size.

NL: Plot dataframe df without a legend.

Target code: `df.plot(legend=False)`

Target action sequence: [Expr, Call, Attribute, Name, df, Load, plot, Load, Reduce, keyword, legend, NameConstant, · · · ]

| Model | Predicted Code | Predicted Action Sequence | $p_{\text{NN}}$ | $p_{k\text{NN}}$ | $\lambda$ | $p(y\|x)$ |
|---|---|---|---|---|---|---|
| ♣ | df.plot(kind='bar', legend=0) | [· · · keyword, kind · · · ] | $p(kind) = 0.63$ 
 $p(legend) = 0.21$ | - | - | $\mathbf{p(kind) = 0.63}$ 
 $p(legend) = 0.21$ |
| ♢ | df.plot(legend=False) | [· · · keyword, legend · · · ] | $p(kind) = 0.63$ 
 $p(legend) = 0.21$ | $p(kind) = 0.12$ 
 $p(legend) = 0.88$ | 0.38 | $p(kind) = 0.44$ 
 $\mathbf{p(legend) = 0.46}$ |
| ♠ | df.plot(legend=False) | [· · · keyword, legend · · · ] | $p(kind) = 0.63$ 
 $p(legend) = 0.21$ | $p(kind) = 0.12$ 
 $p(legend) = 0.88$ | 0.41 | $p(kind) = 0.42$ 
 $\mathbf{p(legend) = 0.48}$ |

NL: Get rid of none values in dictionary kwargs.

Target code: `res = {k: v for k, v in list(kwargs.items()) if v is not None}`

Target action sequence: [Assign, Name, res, Store, Reduce, DictC, Name, k, Load, Name, v, Load, comprehension, · · · ]

| Model | Predicted Code | Predicted Action Sequence | $p_{\text{NN}}$ | $p_{k\text{NN}}$ | $\lambda$ | $p(y\|x)$ |
|---|---|---|---|---|---|---|
| ♣ | kwargs.values() | [Expr, Call · · · ] | $p(Call) = 0.43$ 
 $p(DictC) = 0.38$ 
 $p(Subs) = 0.14$ | - | - | $\mathbf{p(Call) = 0.43}$ 
 $p(DictC) = 0.38$ 
 $p(Subs) = 0.14$ |
| ♢ | {k: v for k, v in list(kwargs.items()) if v is not None} | [Expr, DictC · · · ] | $p(Call) = 0.43$ 
 $p(DictC) = 0.38$ 
 $p(Subs) = 0.14$ | $p(Call) = 0.05$ 
 $p(DictC) = 0.95$ 
 $p(Subs) = 0$ | 0.47 | $p(Call) = 0.25$ 
 $\mathbf{p(DictC) = 0.65}$ 
 $p(Subs) = 0.07$ |
| ♠ | kwargs.items()[0] | [Expr, Subs · · · ] | $p(Call) = 0.43$ 
 $p(DictC) = 0.38$ 
 $p(Subs) = 0.14$ | $p(Call) = 0.21$ 
 $p(DictC) = 0$ 
 $p(Subs) = 0.79$ | 0.36 | $p(Call) = 0.35$ 
 $p(DictC) = 0.14$ 
 $\mathbf{p(Subs) = 0.37}$ |

Table 6: The case study of model canonical incremental adaptation. ✓ represents the correct target code. ♣ is the generated result of BertranX†. ♢ and ♠ are the generated results of updating or not updating the syntactic datastore with 95k external data by $k$NN-TRANX†. The actions enclosed in the red boxes indicate the generated errors. We present the probability distribution of model decisions during the generation of these erroneous tokens.

| Model | Decoding Time (ms/code) |
|---|---|
| BertranX | 357.5($\times$1.00) |
| $k$NN-MT | 1258.4($\times$3.52) |
| $k$NN-TRANX | 722.2($\times$2.02) |

Table 7: Decoding time of different models.

## A.2 Decoding Time

We compared the decoding time of BertranX, $k$NN-MT, and $k$NN-TRANX using the CoNaLa test set. During decoding, the beam size was set to 15. The results are presented in Table 7.