# OpenReview forum: "Syntax-Aware Retrieval Augmented Code Generation"
_EMNLP/2023/Conference — EMNLP 2023 Findings_

### Official Review · Reviewer_Av9a · 2023-08-04

**Soundness:** 3

**Excitement:**

2: Mediocre: This paper makes marginal contributions (vs non-contemporaneous work), so I would rather not see it in the conference.

**Missing References:**

A recent work of CodeT5+ also includes evaluation on retrieval-augmented code generation tasks and should be cited and compared.

**Paper Topic And Main Contributions:**

This paper proposes kNN-TRANX, a token-level retrieval augmented code generation method. kNN-TRANX allows for searches in smaller datastores tailored for the code generation task. It utilizes seq2tree model to get AST information and leverages such syntax constraints for the retrieval of datastores, which reduces the impact of retrieve noise. Results on two public datasets verify the effectiveness of the proposed approach. However, this work does not compare with SoTA retrieval-augmented methods such as REDCODER and the recent CodeT5+ models or more widely adopted code generation benchmarks such as APPS and HumanEval. Besides, the use of seq2tree models is a bit strange as the AST information can be easily obtained via off-the-shelf parsers.

**Questions For The Authors:**

* Why do you need to use seq2tree models to extract AST instead of using a static parser?
* Can you provide comparison with SoTA retrieval-augmented methods such as REDCODER and the recent CodeT5+ models?

**Reasons To Accept:**

* The paper is well written and easy to read.
* The proposed method achieves better results than baselines on two benchmarks. Additional experimental analysis is well conducted to better understand the models.

**Reasons To Reject:**

* One major limitation of this paper is that this work does not compare with SoTA retrieval-augmented methods such as REDCODER and the recent CodeT5+ models or more widely adopted code generation benchmarks such as APPS and HumanEval. These SoTA retrieval-augmented methods often employ an emsembling approach to use a sparse retriever (e.g., BM25) for syntatic matching and a deep retriever (e.g., CodeBERT) for semantic matching. Without comparison to these baselines, it is difficult to determine the technical significance of the proposed model.
* The use of seq2tree models is a bit strange as the AST information can be easily obtained via off-the-shelf parsers.
* The retrieval component of kNN-TRANX is a bit too complicated which might sacrifice its inference efficiency. In Table 5, the authors show some statistics of the training cost, but for code generation systems, the inference efficiency in deployment is often more crucial.

**Reproducibility:**

3: Could reproduce the results with some difficulty. The settings of parameters are underspecified or subjectively determined; the training/evaluation data are not widely available.

**Reviewer Confidence:**

4: Quite sure. I tried to check the important points carefully. It's unlikely, though conceivable, that I missed something that should affect my ratings.

---

> ### Author Rebuttal · Authors · 2023-08-29
>
> Thanks for the comments.
>
> ## Questions for the authors
>
> * **Q1. Why do you need to use seq2tree models to extract AST instead of using a static parser?**
>
>     **A1:** A static parser can easily obtain AST information. However, in our work, the seq2tree model is actually used as a generator rather than a parser. In fact, the static parser is also part of the seq2tree architecture, used to parse the target code into AST. The main functionality of the seq2tree model, as described in section 2.2 of this paper, is to provide a standardized method for generating ASTs using ASDL (To make it easier to understand, we provided a toy example in Figure 2 of the paper to illustrate the generation process of $k$NN-TRANX.). In this way, we can combine program analysis techniques during the generation process to accelerate inference and filter noise.
>
> * **Q2. Can you provide comparison with SoTA retrieval-augmented methods such as REDCODER and the recent CodeT5+ models?**
>
>     **A2:**  We have conducted a detailed evaluation of the performance of four SoTA baselines in the following table.
>
>     |  |  | CoNaLa ||| Django ||
>     | ------ | ------ | ------ | ------ | ------ | ------ | ------ |
>     |  | BLEU | CodeBLEU | EM | BLEU | CodeBLEU | EM |
>     | REDCODER | 35.12 | 36.82 | 6.4 | 75.36 | 73.01 | 78.82 |
>     | CodeT5 base | 36.28 | 35.34 | 6.8 | 76.50 | 71.92 | 80.93 |
>     | CodeT5+ base | 35.84 | 34.76 | 6.4 | 77.32 | 72.28 | 80.98 |
>     | gpt-3.5 3-shots | 36.25 | 38.03 | **7.4** | 68.97 | 72.58 | 58.85 |
>     | Ours | **37.29** | **39.04** | **7.4** | **83.20** | **81.66** | **82.16** |
>
>     As for the APPS and HumanEval datasets, due to limited computational resources, small-scale models struggle to perform well on these two datasets. In future work, we hope to experiment with large language models on these two datasets.
>
> ## Further clarification
>
> * **Inference efficiency:** While we acknowledge the importance of inferring efficiency in deployment, it is true that introducing a retrieval module will unavoidably lead to an increase in inference latency as we discussed in the Limitation part of the paper. However, in our research, we still strive to solve this issue, which can be divided into two parts. Firstly, we have constructed fine-grained datastores based on the kind of AST nodes to narrow down the retrieval space, thereby minimizing the potential impact on latency. This allows for more precise retrieval of information during the code generation process. Secondly, we have ensured that the retrieved target tokens comply with the construction rules of the AST. By doing so, we strike a balance between efficient retrieval and maintaining the structural integrity of the code being generated. While it is true that there may still be some increase in inference latency (shown in Table 7 of the paper), our approach ultimately yields better results in terms of code generation. Additionally, our proposed method offers a more efficient incremental learning capability. This enables the model to have online learning ability without additional training requirements. We believe that these advantages are also significant in real-world applications where efficiency and adaptability are crucial.
>
> * **Missing References:**  We will include more relevant papers in the revised version.
>
> * **Reproducibility concern:** In the event that the paper is accepted, we are committed to providing source code, training parameters, and stored datastores.

---

### Official Review · Reviewer_VgbA · 2023-08-06

**Soundness:** 3

**Excitement:**

3: Ambivalent: It has merits (e.g., it reports state-of-the-art results, the idea is nice), but there are key weaknesses (e.g., it describes incremental work), and it can significantly benefit from another round of revision. However, I won't object to accepting it if my co-reviewers champion it.

**Missing References:**

As mentioned, there are many recent works on LLM-based code generation models. Also, there are other retrieval-augmented code intelligence works such as the following:
Jian Zhang et al., 2020. Retrieval-based neural source code summarization. In Proceedings of the ACM/IEEE 42nd International Conference on Software Engineering (ICSE '20),1385–1397.

**Paper Topic And Main Contributions:**

To reduce extraneous noise in the generation process of pretrained neutral machine translation models, the authors propose kNN-TRANX, a token-level retrieval augmented code generation method. kNN-TRANX enables searches in smaller datastores tailored for the code generation task, leveraging syntax constraints for the retrieval of datastores, thereby mitigating the impact of retrieval noise.

**Questions For The Authors:**

How is the proposed approach compared with large language model (LLM) based code generation methods?

**Reasons To Accept:**

- This paper is generally well written with a good structure. It is easy to follow.

- The idea of combining kNN-MT and Seq2Tree for code generation is intriguing, and the authors proposed three mechanisms to enhance performance: Syntax-constrained token-level retrieval, Meta-k network, and Confidence network.



**Reasons To Reject:**

- Even though the paper has proposed several novel techniques to improve code generation performance, my major concern lies in the lack of comparison with recent large language models (LLMs) such as ChatGPT, StarCoder, and CodeT5+. Existing LLMs have demonstrated significant performance in code generation, and it is suggested that the authors compare their model with these LLMs. At least a discussion on this comparison is required.

- In the Related Work Section, many works on code generation have not been included, especially those LLM-based models.

- In many recent code generation papers, the Pass@K metric is consistently adopted for evaluation. The authors are strongly encouraged to take this metric into consideration during evaluation.

- The core idea of this work is primarily built upon the existing kNN-MT and Seq2Tree models. The authors are encouraged to emphasize the main innovative points and distinct contributions in comparison to these models.

**Reproducibility:**

2: Would be hard pressed to reproduce the results. The contribution depends on data that are simply not available outside the author's institution or consortium; not enough details are provided.

**Reviewer Confidence:**

4: Quite sure. I tried to check the important points carefully. It's unlikely, though conceivable, that I missed something that should affect my ratings.

---

> ### Author Rebuttal · Authors · 2023-08-29
>
> Thank you for your comments.
>
> ## Questions for the authors
>
> * **Q. How is the proposed approach compared with large language model (LLM) based code generation methods?**
>
>     **A:**  To demonstrate the effectiveness of our work, we agree that it would better compare our model with the SoTA code pre-trained models and LLM. We present experimental results that encompass retrieval-based REDCODER, as well as code pre-training models CodeT5 and CodeT5+, alongside the large language model gpt-3.5.
>
>     |  |  | CoNaLa ||| Django ||
>     | ------ | ------ | ------ | ------ | ------ | ------ | ------ |
>     |  | BLEU | CodeBLEU | EM | BLEU | CodeBLEU | EM |
>     | REDCODER | 35.12 | 36.82 | 6.4 | 75.36 | 73.01 | 78.82 |
>     | CodeT5 base | 36.28 | 35.34 | 6.8 | 76.50 | 71.92 | 80.93 |
>     | CodeT5+ base | 35.84 | 34.76 | 6.4 | 77.32 | 72.28 | 80.98 |
>     | gpt-3.5 3-shots | 36.25 | 38.03 | **7.4** | 68.97 | 72.58 | 58.85 |
>     | Ours | **37.29** | **39.04** | **7.4** | **83.20** | **81.66** | **82.16** |
>
> 	One can see that our model performs better on three evaluation metrics. Furthermore, it is important to note that our method can be integrated with existing language models.
>
> ## Further clarification
>
> * **Related works to be added:** In the era of large language models, we fully understand the necessity of relevant discussions. We will include additional research on large language models in the revised version of the paper.
>
> * **Metrics**: We acknowledge the effectiveness of using pass@k as an indicator for assessing code correctness. However, these two datasets do not provide standardized variable names and other constants, so the pass@k metric cannot be effectively applied. As for the datasets that can be evaluated using the pass@k metric, due to limited computational resources, small models struggle to generate high-quality method-level code. Hence, for our experiments on these two code datasets, we performed evaluations at the line level. In addition to BLEU and EM, which have been used in previous studies, we have introduced the CodeBLEU metric in our paper and presented CodeBERTScore as an automatic evaluation metric in the following table, showcasing the efficacy of our approach.
>
>     |  | CoNaLa |||| Django ||||
>     | ------ | ------ | ------ | ------ | ------ | ------ | ------ | ------ | ------ |
> 	|  | $P$ | $R$ | $F_{1}$ | $F_{3}$ | $P$ | $R$ | $F_{1}$ | $F_{3}$ |
>     | TRANX | .811 | .796 | .802 | .797 | .915 | .902 | .905 | .908 |
>     | Reranker | .823 | .798 | .809 | .800 | .932 | .925 | .932 | .930 |
>     | Ext-codegen | .809 | .805 | .806 | .805 | .935 | .931 | .933 | .932 |
>     | TAE | .812 | .808 | .804 | .803 | .941 | .934 | .936 | .935 |
>     | BertranX | .821 | .812 | .815 | .812 | .987 | .982 | .982 | .981 |
>     | REDCODER | .832 | .825 | .822 | .818 | .986 | .983 | .985 | .981 |
>     | CodeT5 base | .831 | .823 | .826 | .824 | .985 | .982 | .984 | .983 |
>     | CodeT5+ base | .831 | .815 | .822 | .818 | .986 | .981 | .986 | .982 |
>     | gpt-3.5 3-shots | .838 | **.835** | .826 | .827 | .974 | .972 | .974 | .969 |
>     | Ours | **.842** | .827 | **.833** | **.828** | **.989** | **.985** | **.987** | **.985** |
>
> * **Novelty:** The success of $k$NN-MT in enhancing text generation models and its ease of model updating surprised us. However, despite its potential benefits, this token-level retrieval-enhanced model is not widely adopted in practice. The primary concern is the increasing inference latency and additional storage resources. Furthermore, retrieval-based approaches introduce noise that can negatively impact generation, a critical issue in code generation tasks where a single inappropriate token may lead to the failure of an entire code block. In contrast to traditional text generation tasks, our work relies on domain knowledge and program analysis techniques. By leveraging these techniques, we construct multiple datastores that encompass both syntax and semantics information. This approach improves retrieval efficiency and reduces noise through restrictive retrieval. Additionally, our model introduces a self-adaptive combination method that significantly enhances the effectiveness of model generation. We have also investigated the impact of domain knowledge on model generation and devised a convenient approach for model updating, preserving competitive performance while minimizing additional storage consumption.
>
> * **Reproducibility concern:** In the event that the paper is accepted, we are committed to providing source code, training parameters, and stored datastores.
>
> * **Missing References:**  We will include more relevant papers in the revised version.

---

### Official Review · Reviewer_dTew · 2023-08-09

**Typos Grammar Style And Presentation Improvements:** 1. The datastore construction equatio…
**Soundness:** 4

**Excitement:**

4: Strong: This paper deepens the understanding of some phenomenon or lowers the barriers to an existing research direction.

**Missing References:**

https://arxiv.org/abs/2201.11227 -- introduces constraint guided generation

**Paper Topic And Main Contributions:**

The paper explores the problem of improving the quality of language guided code generation using token level retrieval from an offline datastore that is separately constructed at train time. The paper improves upon this idea of token level retrieval by making the following contributions:
(1) Instead of directly generating code, the paper generates AST which allows the system to constrain generation and improve quality in terms of generating more syntactic code.
(2) The paper extends the meta-K networks for choosing the optimal retrieval cache size.
(3) Instead of using a fixed confidence threshold (as done in previous works), the paper employs a confidence network for combining the predictions from the neural and retrieval models.

The paper improves traditional code generation via token level retrieval which is particularly useful in low-resource domains where training data is scarce.

**Questions For The Authors:**

- A. Why was BertanX chosen as the base model for this work on not a more recent NMT system like T5, BART.
- B. Why are BLEU and CodeBLEU missing in Table 2 for Reranker?
- C. Do the authors know how the performance might vary based on the underlying NMT model? Are there specific domains or tasks where this can be more impactful? maybe low-resource languages?

**Reasons To Accept:**

The strengths of this paper are:
1. Adapting token level retrieval for generation tasks to the domain of code generation and discussing the insights and challenges.
2. Well written and easy to follow. It is a simple system that works well for the domain.
3. Improving over existing retrieval augmented generation systems with meta-K networks and confidence networks.

**Reasons To Reject:**

The paper proposes a very nice system for improving code generation performance using token level retrieval but it has some weaknesses that need to be addressed.
1. The idea around token level retrieval for improved augmentation in itself is not novel and has been extensively used in NLP.
2. The novel components in BertranX seem to be the separate data store, constrained generation and the confidence network for combining model and retrieval predictions. It is not clear how significant these are as the ablations are not very indicative. Perhaps it would be more convincing if the authors test this on a low resource programming language which might benefit from retrieval.
3. The evaluation is not very thorough. Some suggestions
    - There needs to be some comparison with SOTA code generation models like T5, Llama2, GPT-3.5
    - Perhaps, Conala and Django are not the best languages to test this on as a lot of models have seen these in their pretrasining.
    - The metrics reported are standard for code generation but specifically for this work, it would be interesting to see the overall syntactic generation fraction (since the authors claim that they generate more syntactic code) and CodeBERTScore since Conala is known to be a noisy dataset with no normalization for variable names and other constants.

**Reproducibility:**

4: Could mostly reproduce the results, but there may be some variation because of sample variance or minor variations in their interpretation of the protocol or method.

**Reviewer Confidence:**

4: Quite sure. I tried to check the important points carefully. It's unlikely, though conceivable, that I missed something that should affect my ratings.

---

> ### Author Rebuttal · Authors · 2023-08-29
>
> Thanks for the comments.
>
> ## Questions for the authors
>
> * **Q1. Why was BertanX chosen as the base model for this work on not a more recent NMT system like T5, BART.**
>
> 	**A1:** Our approach aims to address the token-level retrieval challenge by leveraging the AST feature of programming languages. To achieve this, we adopt the BertranX seq2tree model as the basis. However, it should be emphasized that our approach can also be applied to seq2seq models. One viable option is to enhance the vocabulary table of conventional seq2seq models (such as T5 and BART) by incorporating the actions required for constructing ASTs, which would equip these models with comparable capabilities.
>
> * **Q2. Why are BLEU and CodeBLEU missing in Table 2 for Reranker?**
>
>     **A2:** To ensure a comprehensive comparison using more evaluation metrics, we tried to replicate the results by utilizing the checkpoints provided by the authors. However, the checkpoint for the Reranker on the Django dataset was unavailable, so we had to rely on the results from the original paper for this particular component. Nevertheless, we used their method to reproduce the results on the Django dataset, which are shown below.
>
> 	|  |  | Django | |
> 	| ------ | ------ | ------ | ------ |
> 	|  | BLEU | CodeBLEU | EM |
> 	| Reranker | 73.26 | 71.29 | 80.18 |
> 	| Ours | **83.20** | **81.66** | **82.16** |
>
> * **Q3. Do the authors know how the performance might vary based on the underlying NMT model? Are there specific domains or tasks where this can be more impactful? maybe low-resource languages?**
>
> 	**A3:** Although the underlying model can indeed impact performance, prior research indicates that powerful models can still reap benefits from token-level retrieval. Furthermore, by creating multiple domain-specific datastores using the target programming language, we can achieve domain adaptation by seamlessly switching between these datastores. For low-resource languages, they are often poorly optimized in the model, so we believe that introducing external knowledge will be helpful for the model to generate these languages.
>
> ## Further clarification
>
> * **Novelty:** We agree that token-level retrieval methods have shown excellent performance in the field of NLP, especially in the NLG domain. However, our method aims to generate code written in programming languages, which is different from natural language. Programming languages need to meet strict syntactic constraints (otherwise the code would not compile), which traditional text generation models struggle to ensure (as shown in Table 1 of the paper). In addition, retrieval-based generation methods require higher inference latency and additional storage, which hinders the development of this approach. To address these issues, we leverage domain knowledge and program analysis techniques to ensure the syntactic correctness of the generated code while improving inference efficiency and reducing storage consumption.
>
> * **Ablation study:** While the ablation experiments conducted so far may not fully offer conclusive evidence about the significance of the components we propose, they indeed show a substantial impact on evaluation metrics. It is intriguing to consider testing this concept in a low-resource programming language, as it would provide valuable insights. Notably, as illustrated in Section 4.3, we simulated an environment with limited resources and the experimental results convincingly demonstrate that the model still benefits from retrieval.
>
> * **Evaluation:**
>
>    * **Add baselines**: To further demonstrate the effectiveness of our approach, it would better compare our model with the SoTA code pre-trained models and LLM. Based on the suggestions of the reviewers, we added four baselines for comparison. The experimental results are presented below.
>
>     |  |  | CoNaLa ||| Django ||
>     | ------ | ------ | ------ | ------ | ------ | ------ | ------ |
>     |  | BLEU | CodeBLEU | EM | BLEU | CodeBLEU | EM |
>     | REDCODER | 35.12 | 36.82 | 6.4 | 75.36 | 73.01 | 78.82 |
>     | CodeT5 base | 36.28 | 35.34 | 6.8 | 76.50 | 71.92 | 80.93 |
>     | CodeT5+ base | 35.84 | 34.76 | 6.4 | 77.32 | 72.28 | 80.98 |
>     | gpt-3.5 3-shots | 36.25 | 38.03 | **7.4** | 68.97 | 72.58 | 58.85 |
>     | Ours | **37.29** | **39.04** | **7.4** | **83.20** | **81.66** | **82.16** |
>
> 	One can see that our model performs better on three evaluation metrics. Furthermore, it is important to note that our method can be integrated with existing models.
>
>     * **Training data leakage**: Our base model utilizes BERT as the encoder and a decoder that is randomly initialized. Compared to pre-trained encoder-decoder and decoder-only architectures, we believe that our model has a lower risk of information leakages.
>
> 	* **Evaluation metrics:** To demonstrate that our model can generate code with more accurate syntax, we provide the syntax match score, which reflects the degree of syntax matching in code.
>
> 	| Model | TRANX | Reranker | Ext-codegen | TAE | BertranX | REDCODER | CodeT5 base | CodeT5+ base | gpt-3.5 3-shots| Ours|
>     | ------ | ------ | ------ | ------ | ------ | ------ | ------ | ------ | ------ | ------ | ------ |
> 	|Syntax Match| .287 | .284 | .322| .318 | .347 | .344 | .339 | .341 | .357 | **.368** |
>
> 	CodeBERTScore is also a widely adopted automated code evaluation metric, and we provide its evaluation below.
>
>     |  | CoNaLa |||| Django ||||
>     | ------ | ------ | ------ | ------ | ------ | ------ | ------ | ------ | ------ |
> 	|  | $P$ | $R$ | $F_{1}$ | $F_{3}$ | $P$ | $R$ | $F_{1}$ | $F_{3}$ |
>     | TRANX | .811 | .796 | .802 | .797 | .915 | .902 | .905 | .908 |
>     | Reranker | .823 | .798 | .809 | .800 | .932 | .925 | .932 | .930 |
>     | Ext-codegen | .809 | .805 | .806 | .805 | .935 | .931 | .933 | .932 |
>     | TAE | .812 | .808 | .804 | .803 | .941 | .934 | .936 | .935 |
>     | BertranX | .821 | .812 | .815 | .812 | .987 | .982 | .982 | .981 |
>     | REDCODER | .832 | .825 | .822 | .818 | .986 | .983 | .985 | .981 |
>     | CodeT5 base | .831 | .823 | .826 | .824 | .985 | .982 | .984 | .983 |
>     | CodeT5+ base | .831 | .815 | .822 | .818 | .986 | .981 | .986 | .982 |
>     | gpt-3.5 3-shots | .838 | **.835** | .826 | .827 | .974 | .972 | .974 | .969 |
>     | Ours | **.842** | .827 | **.833** | **.828** | **.989** | **.985** | **.987** | **.985** |

---

### Meta-Review · Area_Chair_44a8 · 2023-09-19

**Recommendation:** 4

**Metareview:**

The paper investigates methods for improving the performance of token-level retrieval augmented models for code generation.

The proposed framework seems to work well and achieves good results, compared to the chosen baselines.
All reviewers comment that the paper is very well written and easy to read.

Reviewers highlighted concerns about lack of comparison to state-of-the-art models and LLMs. Authors have provided some additional results in the comments which seem to demonstrate the proposed method outperforms several LLMs.
Reviewers also voiced concerns about the novelty and motivation of the method.
The evaluation could be improved, with better ablation tests to clearly show whether the proposed components are responsible for performance improvements.

---

### Decision · Program_Chairs · 2023-10-07

**Decision:**

Accept-Findings

**Comment:**

The paper investigates methods for improving the performance of token-level retrieval augmented models for code generation.

The proposed framework seems to work well and achieves good results, compared to the chosen baselines.
All reviewers comment that the paper is very well written and easy to read.

Reviewers highlighted concerns about lack of comparison to state-of-the-art models and LLMs. Authors have provided some additional results in the comments which seem to demonstrate the proposed method outperforms several LLMs.
Reviewers also voiced concerns about the novelty and motivation of the method.
The evaluation could be improved, with better ablation tests to clearly show whether the proposed components are responsible for performance improvements.